# Explainable AI decision support improves accuracy during telehealth strep throat screening

Catalina Gomez [1], Brittany-Lee Smith[2], Alisa Zayas[2], Mathias Unberath [1,2,4] & Therese Canares [3,4]

## Abstract

**Background** Artificial intelligence-based (AI) clinical decision support systems (CDSS) using unconventional data, like smartphone-acquired images, promise transformational opportunities for telehealth; including remote diagnosis. Although such solutions' potential remains largely untapped, providers' trust and understanding are vital for effective adoption. This study examines how different human–AI interaction paradigms affect clinicians' responses to an emerging AI CDSS for streptococcal pharyngitis (strep throat) detection from smartphone throat images.

**Methods** In a randomized experiment, we tested explainable AI strategies using three AI-based CDSS prototypes for strep throat prediction. Participants received clinical vignettes via an online survey to predict the disease state and offer clinical recommendations. The first set included a validated CDSS prediction (Modified Centor Score) and the second introduced an explainable AI prototype randomly. We used linear models to assess explainable AI's effect on clinicians' accuracy, confirmatory testing rates, and perceived trust and understanding of the CDSS.

**Results** The study, involving 121 telehealth providers, shows that compared to using the Centor Score, AI-based CDSS can improve clinicians' predictions. Despite higher agreement with AI, participants report lower trust in its advice than in the Centor Score, leading to more requests for in-person confirmatory testing.

**Conclusions** Effectively integrating AI is crucial in the telehealth-based diagnosis of infectious diseases, given the implications of antibiotic over-prescriptions. We demonstrate that AI-based CDSS can improve the accuracy of remote strep throat screening yet underscores the necessity to enhance human–machine collaboration, particularly in trust and intelligibility. This ensures providers and patients can capitalize on AI interventions and smartphones for virtual healthcare.

## Plain language summary

Strep pharyngitis, or strep throat, is a bacterial infection that can cause a sore throat. Artificial intelligence (AI) can use photos taken on a person's phone to help diagnose strep throat, offering an additional way for doctors to screen patients during virtual appointments. However, it is currently unclear whether doctors will trust AI recommendations or how they might use them in decision-making. We surveyed clinicians about their use of an AI system for strep throat screening with smartphone images. We compared different ways of providing AI recommendations to standard medical guidelines. We found that all tested AI methods helped clinicians to identify strep throat cases. However, clinicians trusted AI less than their usual clinical guidelines, leading to more requests for follow-up in-person testing. Our results show how AI may improve the accuracy of pharyngitis assessment. Still, further research is needed to ensure doctors trust and collaborate with AI to improve remote healthcare.

Artificial Intelligence (AI) plays a major role in the implementation of telehealth technologies and delivery of virtual health care[1]. Emergent trends that involve AI include patient monitoring, healthcare information technology, intelligent assistance diagnosis, and information analysis collaboration[2,3], leveraging conventional patient data found in electronic medical records. Dermatology is an example of a well-known existing area for tele-diagnosis, where automated algorithms have achieved diagnostic accuracy close to human experts[4] and human–machine partnership has

been evaluated[5]. Identifying other applications where AI can improve the telehealth experience is crucial for extending health access in remote scenarios[6]. One particularly promising opportunity is to support remote and asynchronous diagnosis of diseases by combining the analytic power of AI algorithms with the wealth of alternative data sources provided by novel and accessible technologies, such as smartphones.

During telehealth visits, providers rarely have the opportunity to support clinical decisions with point-of-care laboratory tests. AI algorithms

[1]Department of Computer Science, Johns Hopkins University, Baltimore, MD, USA. [2]Johns Hopkins University School of Medicine, Baltimore, MD, USA. [3]Division of Pediatric Emergency Medicine, Johns Hopkins University School of Medicine, Baltimore, MD, USA. [4]These authors jointly supervised this work: Mathias Unberath, Therese Canares. ✉e-mail: unberath@jhu.edu

combined with available data sources, such as smartphone images, offer a solution[7,8]. These approaches usually rely on complex image recognition algorithms, such as Convolutional Neural Networks, that can achieve certain performance requirements but have unclear inner working mechanisms and may not satisfy end users' needs[9]. For providers to act on recommendations from AI-based clinical decision support systems (CDSS), it is hypothesized that it is necessary to invoke trust and intelligibility by augmenting AI models to justify the validity of their recommendation[10–12]. Therefore, to leverage the potential of AI-based diagnostic testing for telehealth, it is paramount to gain a clear understanding of how clinicians perceive these systems and which design choices affect trust, transparency, usability, and other human factors.

In this work, we contextualize the study of AI as a CDSS in the telehealth domain for the prediction of streptococcal pharyngitis (strep throat) from images of the throat, acquired with a smartphone[13,14]. Under this assistance scenario, both the clinician and AI algorithm review an image, the main source for the diagnostic decision. Remote review of smartphone images is not a standard of care currently, nor is it consistently performed with mobile or remote monitoring devices beyond conventional clinical settings. Importantly, patients should provide adequate photos for both algorithmic and human processing, which can be achieved with additional guidance, feedback, and verification. In particular, clinicians have no unified practice in evaluating pharyngitis through telehealth, providing an opportunity to study interpretations and effectiveness for such an AI CDSS to be integrated and expand diagnostic capacities outside the clinic. The standard of care to distinguish strep throat from viral pharyngitis includes a throat culture or rapid antigen test. This guides clinicians to appropriately prescribe antibiotic treatment to decrease the severity of symptoms, risk of transmission, and further complications[15]. Therefore, patients experiencing a sore throat must currently seek clinical evaluation for a diagnostic test and to obtain the appropriate treatment. Meanwhile, there are other protocols where it is common practice to diagnose strep throat clinically without sending throat cultures. Because an evaluation through telehealth services generally lacks point-of-care diagnostic tests, telehealth in its current form is a suboptimal option for patients with pharyngitis symptoms. Clinical prediction rules have been developed to estimate the probability of strep throat, and are best used to guide whether a diagnostic test is indicated. A simple and commonly accepted rule, known as the Centor Score, considers four signs and symptoms to provide an additive score associated with the likelihood of strep throat; the Modified Centor score includes age as a risk factor[16,17], unlike other rules more recently proposed, such as Fever-PAIN[18]. Even though the diagnostic process combines clinical findings and the clinician's own judgment, physicians' accuracy at identifying strep throat is slightly above 50%[17,19].

To better understand how AI systems may be adopted in healthcare, previous efforts have been concentrated on understanding the context and evaluating whether certain human-engineering goals are fulfilled[20]. To achieve the former, formative research techniques[21] and co-design strategies with end users[22–25] have been followed in diverse medical applications to inform design choices of transparent algorithms and what people expect from interacting with them. Despite having a large collection of explainable AI techniques developed or adapted to healthcare applications[26–30], whether an algorithm affords transparency is not purely a computational problem[31,32]. In domains such as breast imaging and dermatology, where deep learning models have been well-established and extensively studied[5,33], there exists a notable interest in empirical evaluations of explainability techniques that can be directly incorporated into existing models. The evaluation of AI interface designs includes information that supports predictions in different formats, such as highlighting or summarizing relevant features[34–38] and presenting similar cases[39–41]. In parallel, certain studies have chosen to introduce explainability features into interfaces all at once[42–44], which can lead to convoluted user experiences, thereby hindering the capacity to evaluate their efficacy or pinpoint which components are genuinely beneficial. In less explored domains, there is a surge in innovating explanation methodologies utilizing mock-ups to pilot novel concepts,

as evidenced in tasks such as antidepressant[38] and chemotherapy prescriptions[45], or automated biosignal analysis[46,47]. While previous experimental studies have explored common explainable AI strategies to support clinical decision-making tasks that range from disease diagnosis to medication, the findings are context-dependent and sometimes conflicting, particularly regarding the usefulness of explanations. This highlights the relevance of following human-centered design principles in novel tasks within the domain of telehealth to inform the development of AI algorithms that can potentially support such tasks. In the context of virtual healthcare, a few works have studied the overall user experience using telehealth services[6], the interpretation of rich patient information remotely[48], and the context-dependent nature of explanations for disease self-management in non-experts with bots[49]. We must meticulously conceptualize AI's role in the remote screening of infectious diseases from non-conventional image sources, exploring essential assistance elements to foster successful collaborations with the envisioned stakeholders. Our work explores human factors and the role of explainable AI in a CDSS to achieve desired interactions with providers during the remote screening for strep throat, combining the potential of AI algorithms and smartphones. Accurate assessments and treatment recommendations are of paramount importance, not only to provide optimal care but also for effective utilization of telehealth services, e.g., avoiding inconvenient or unnecessary in-person healthcare visits.

Due to the challenges associated with screening for strep throat in settings without point-of-care tests and to support clinicians operating without medical tests, the aim of this study was to evaluate the use of an AI-based CDSS during telehealth services to identify strep throat from images of the throat. In particular, our focus was on the influence of using alternate explainable AI elements in a CDSS on clinicians' accuracy in predicting strep throat and which transparency techniques are associated with improved trust and understanding of the AI system. In the formulation of such AI systems, we implemented core practices of human–computer interaction (HCI) design, approaching end users and developing a mock-up AI system as an early-stage interactive AI prototype[22,50]. This approach allows us to rapidly explore specific design opportunities and probe user behaviors, without compromising resources in the implementation of a functioning AI system that may not achieve certain human factors goals. The mock-ups feature various techniques the AI system might provide to support its predictions, including the presentation of algorithmic performance, a method that presents key visual characteristics of throat images, and one that displays examples of similar images previously analyzed by the system. Our findings from an online study demonstrate the potential of AI-based CDSS to support strep throat prediction as clinicians' accuracy improves compared to using the Centor Score. However, participants exhibit lower trust in AI advice both in their subjective perception and choice of follow-up suggestions. Furthermore, we identify trade-offs between trust and understanding achieved with different explainable AI elements.

## Methods
### Study design
We formulated a user study to evaluate an AI-based CDSS that assists strep pharyngitis screening on telehealth. We defined a randomized three-arm comparison experiment, where each arm corresponds to an explainable AI element. Participants started by completing the diagnostic task using their clinical judgment supported by the Modified Centor score criteria (hereafter referred to as Centor Score). Then, they were randomly assigned to one of the three arms in which AI predictions were augmented with information that can be understandable to the end users, as in a between-subjects design. The explainable AI elements we evaluated addressed the interpretability of performance outcomes and understanding of model internals by summarizing features or selecting instances of the dataset[32]. Figure 1 presents the allocation of participants and assessments to be completed during the study. In particular, we hypothesize that an AI system that presents information that is understandable to clinicians will promote trusting behaviors in users towards incorporating AI advice[51] and will improve team performance

compared to the commonly used clinical rule. More specifically, we expect improved yet different behaviors and perceptions based on the elements that provide transparency to the AI's prediction.

### AI-based CDSS prototype design

To ground the choice of information that is potentially relevant for explaining or supporting an AI-based prediction of strep throat, we considered the context and task to be completed with the CDSS. The Modified Centor Score (CS) was developed to assist physicians to estimate the probability of strep throat[15,52] using the following criteria: age, tonsillar exudate, swollen tender anterior cervical nodes, history of fever, and absence of cough. Within our multidisciplinary team of computer scientists, clinicians, and user interface designers, the ideation process was based on current clinical practice and transparency techniques previously used in medical tasks[31] to formulate the following transparency elements to support AI predictions:

**Interpretation of performance characteristics (PC).** As clinicians interpret diagnostic test outcomes considering known measures of validity, such as sensitivity and specificity, we presented the AI predictions along with some model's performance metrics[53]. Evaluating the AI model on held-out data (i.e., data not used for model training) using task-related metrics is part of the algorithm pipeline to ensure that the model performs well not just on the data it was trained on, but also on new, unseen data. Such indicators of the model's true performance can be used to communicate the stated performance, usually on a test set, before deployment where novel instances may be encountered[54].

**Explain by Feature Contribution (EFC).** We identified key signs of a bacterial infection that can be derived from throat images, e.g., exudates or swelling of tonsils and tongue appearance. Then, we selected an explainability technique that highlights features considered most important by the algorithm in its prediction in a post-hoc fashion, i.e., after the prediction has been generated[55]. In practice, by annotating these signs on throat images, their influence can be computed in a trained classifier that predicts positive or negative strep cases, using tools like Shapley values for example[56].

**Explain by Examples (EE).** Clinical reasoning might incorporate case experience into the analysis of the current patient information through comparisons with similar cases[32]. As clinicians are used to thinking of examples or analogies to diagnose, we included similar instances to the current case that the AI had analyzed and had the same prediction, as well as other instances that received a different diagnosis for comparison purposes.

To design the user interface, we created mock-ups of the AI algorithm by manually defining its outputs. The AI-based diagnostic advice corresponded to a prediction of either a positive or negative diagnosis for strep throat. To reflect the AI model's diagnostic accuracy for strep throat, we reported metrics related to its discriminative properties (sensitivity and specificity) and predictive abilities (positive and negative predictive values) next to the corresponding values of the rapid strep test for reference. We displayed the performance characteristics of both the AI-based CDSS and strep test in all the cases to be diagnosed, as illustrated in Fig. 2. The values of the former were defined in the range of what is expected to be clinically acceptable based on preceding interviews with clinicians and the latter were adopted from existing reports of rapid antigen detection tests or throat cultures that correspond to the images[57]. To present feature-based explanations, we provided bar plots showing the individual contribution of four features to the prediction (strawberry tongue, redness, pus, and enlarged tonsils), as illustrated in Fig. 2. The direction of the bar (above or below zero) indicates the support towards a positive or negative prediction. In the example-based explanations, we presented four examples including two cases with similar exam findings to the current case, e.g., with positive strep outcomes if the current case had a positive strep throat prediction. We manually selected similar cases based on the visual appearance of each case. The other two cases corresponded to examples with the opposite prediction to that of the current case. Since there is no similarity requirement, we used the same two positive or negative instances for negative and positive predictions of the current case, respectively. The user interface with the Centor score is shown in Fig. 2 including the rules, score, and likelihood of strep throat. Given the limited number of cases, the AI predictions were always correct, i.e., positive for confirmed cases with a throat culture and negative otherwise. This translates to an observed accuracy of 100% during the study; however, participants were not aware of this, and we did not communicate performance-related information given that one of our experimental manipulations involves the interpretation of performance characteristics. Likewise, the Centor score suggested a higher likelihood (11–17%) of strep pharyngitis for the positive cases and a lower likelihood (5–10%) for negative cases, indicating the level of risk of strep throat.

### Reporting summary

Further information on research design is available in the Nature Portfolio Reporting Summary linked to this article.

### Procedure

Participants who satisfied the inclusion criteria (details in the section "Participants") and agreed to join the study after giving their informed consent proceeded to complete eight diagnostic tasks. The vignette of each case included the patient's age and sex, their symptoms, and an image of their throat. To keep the study duration short, participants only evaluated four cases for each condition (two positive and two negative cases of strep throat). We selected eight cases from a cross-sectional study conducted at urban, tertiary care adult, and pediatric emergency departments; four of which were assigned to the Centor score group, and the other four to the

**Fig. 1 | Flowchart of participant allocation and study task completion.** Participant allocation and task completion flowchart in the study. XAI-1–XAI-3 represents the different explainable AI (XAI) elements. Each case corresponds to a clinical vignette for strep pharyngitis screening.

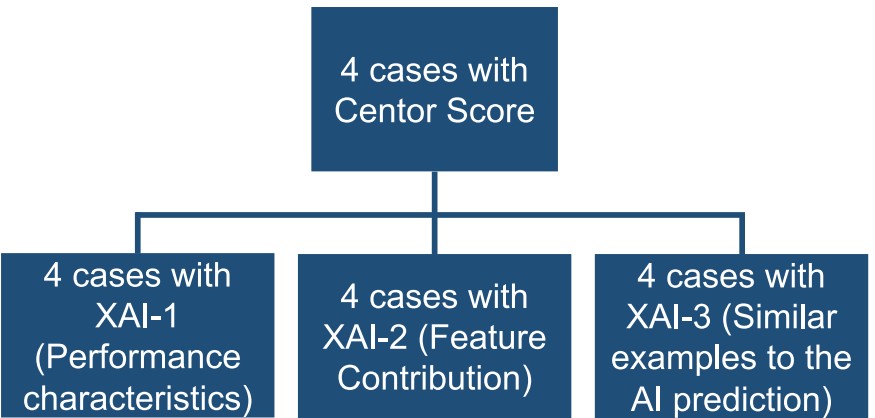

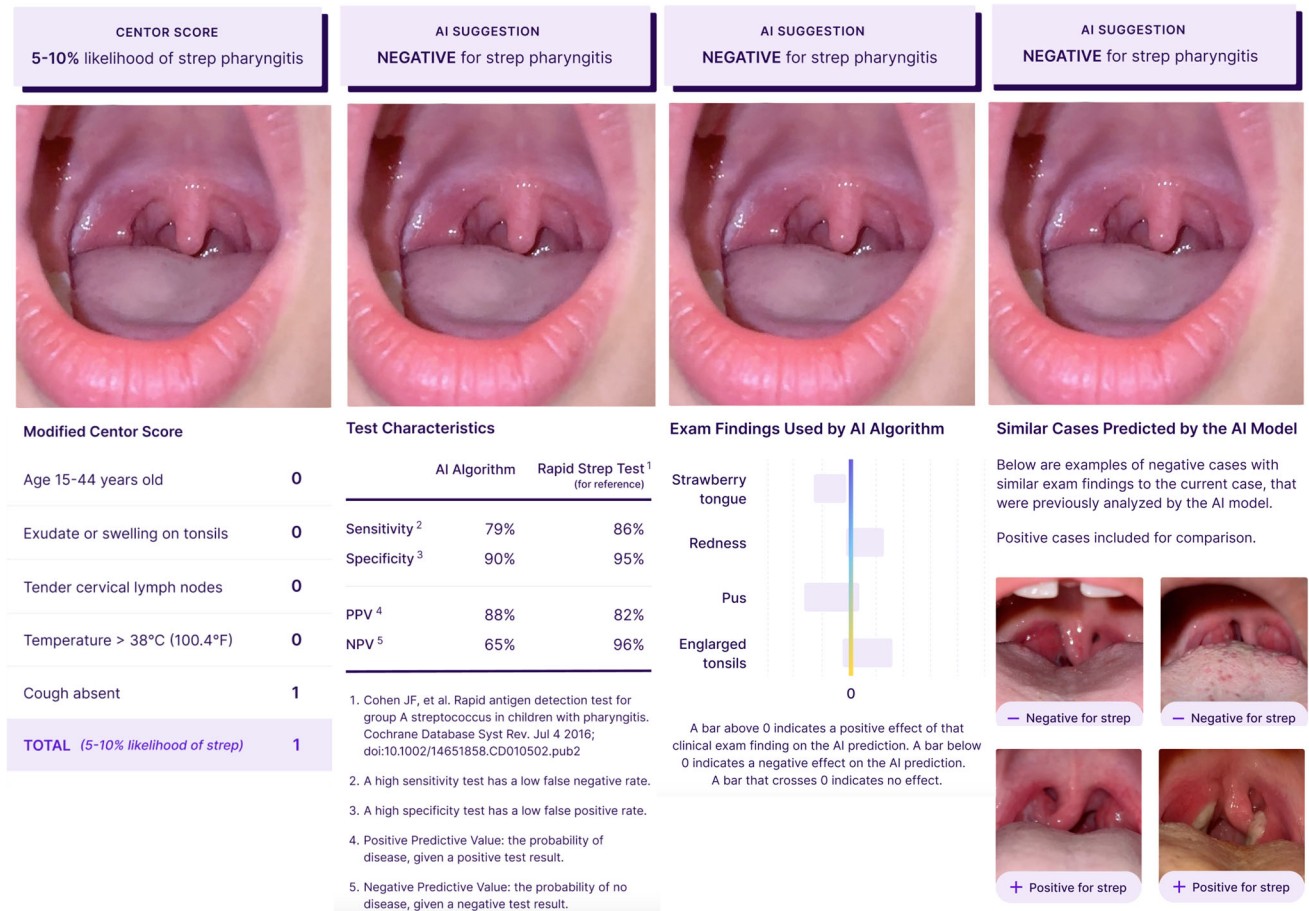

**Fig. 2 | User interface design and experimental conditions for CDSS types.** Visualization of the user interface for strep throat screening using different types of support. From left to right: CDSS with the Centor Score, XAI-1: AI-based CDSS with performance characteristics (PC), XAI-2: AI-based CDSS with Explain by Feature Contribution (EFC), and AI-3: AI-based CDSS with explain by examples (EE).

**Table 1 | Prediction and case management questions in the main study task**

| Survey question | Answer options |
| --- | --- |
| Based on the data provided, I predict this patient is: | Negative for streptococcal pharyngitis |
| | Positive for streptococcal pharyngitis |
| Based on the data provided, if this was a telehealth visit, I would: | Advise supportive care |
| | Give a prescription of antibiotics for bacterial pharyngitis |
| | Advise the patient to get a rapid antigen test/throat culture |

Survey questions at each patient vignette after considering the information from the CDSS. The first question corresponds to the prediction of the case presented and the second one to how to manage the case. Participants had to select one of the answer options.

AI-based CDSS with one type of support information. Among the positive and negative cases, the difficulty level varied by how misleading the case might be, i.e., a difficult case presents visual cues that suggest an incorrect diagnosis while an easy case presents common cues representative of the correct diagnosis. The first four cases contained the Centor score criteria, and then, participants were randomized into one of the three AI transparency groups (PC, EFC, or EE) using the REDCap randomization module in blocks of 12. The four cases in the treatment group displayed the AI's predictions and the corresponding supporting information. Participants had to determine whether each case was positive or negative for strep throat and choose among three potential follow-up actions, as shown in Table 1. These included advising supporting care, treatment with antibiotics, or a referral for a strep test. In addition to measuring diagnostic decisions, this additional question allowed us to assess how participants would manage sick patients who actually need antibiotic treatment and cases to be treated with symptomatic relief. After evaluating each case, participants rated their agreement with three trust-related statements and three statements related to their understanding of the CDSS on a 5-point Likert Scale (listed in Supplementary Methods). The questions were derived from surveys on explainable AI and human–computer trust. The study was conducted online using REDCap electronic data capture tools hosted at Johns Hopkins University[58,59] and was approved by the Johns Hopkins Medicine Internal Review Board (IRB00277755).

## Outcome measures

Based on the participant's strep prediction and the ground truth of each case, we compute the sensitivity and specificity, positive and negative predictive value (PPV and NPV, respectively) under each condition to report the task

performance. The range of the values we report is between [0, 1]. We quantify the agreement with the CDSS as the fraction of cases in which the user's final response matched the prediction provided by the AI or the strep likelihood from Centor score, i.e., a positive diagnosis with higher likelihood values. To assess participants' perceived trust in the CDSS, they rated their agreement with three trust-related statements on a 7-point Likert Scale (Cronbach's $\alpha = 0.95$). As an indirect measure of how much participants trust the CDSS, we report the fraction of negative cases in which participants suggested a confirmatory test. Participants rated their subjective understanding of the CDSS through three statements on a 7-point Likert scale (Cronbach's $\alpha = 0.79$). See Supplementary Method for a complete list of the statements.

## Power analysis

We conducted an a priori data sample estimation following the reported statistics in a study design that compares participants, an AI model, and the team performance to detect malignant skin lesions[60]. From informed

**Table 2 | Sensitivity and specificity analysis across CDSS**

| Measure | Predictors | Estimate | Standard error | t | p-value |
|---|---|---|---|---|---|
| Sensitivity | Intercept | 0.07 | 0.04 | 2.04 | 0.043 |
| | PC | 0.52 | 0.04 | 14.14 | $<2.0 \times 10^{-16}$ |
| | EFC | 0.51 | 0.04 | 13.82 | $<2.0 \times 10^{-16}$ |
| | EE | 0.52 | 0.04 | 14.85 | $<2.0 \times 10^{-16}$ |
| Specificity | Intercept | 0.80 | 0.05 | 16.53 | $<2.0 \times 10^{-16}$ |
| | PC | 0.02 | 0.05 | 0.37 | 0.711 |
| | EFC | −0.09 | 0.05 | −1.80 | 0.073 |
| | EE | 0.15 | 0.05 | 2.98 | 0.003 |

Linear models for sensitivity and specificity measures. Centor score was the reference level. Random effects in the sensitivity model: $\sigma^2 = 0.004$, $N_{ID} = 121$, observations = 242, marginal $R^2 = 0.623$, conditional $R^2 = 0.657$. The linear model for specificity did not include random effects, adjusted $R^2 = 0.053$.

guesses for four experimental groups and to achieve at least 0.8 power, we calculated a total sample size of 76 participants using GPower software.

## Participants

In total, we recruited 295 participants via online distribution of our survey. The inclusion criteria were primary care providers, including physicians, nurse providers, and physician assistants, who practice urgent care or primary care telehealth in the United States. We excluded participants who reported their main use of telehealth was for mental health, surgical or subspecialty care. After considering an agreement to participate in the study through informed consent, 126 participants met the inclusion criteria for the study. For the data analysis, we included 121 valid participants (90 female and 31 male) after filtering incomplete responses and participants who did not identify a single positive strep case, as this may indicate a lack of task attention. Supplementary Fig. 1 presents the enrollment of participants in detail. The sample included 48 physicians, 13 nurse practitioners, and 60 physician assistants. The years of clinical experience ranged from 1 to 3 ($n = 22$), 4–10 ($n = 42$), 11–15 ($n = 23$), 16–20 ($n = 10$), 20+ ($n = 24$). Supplementary Table 1 summarizes the demographic information of the participants sample.

## Statistics and reproducibility

We used mixed-effects regression models to analyze the relationship between each continuous variable and the type of support in the CDSS compared to the clinical prediction rule. The models included fixed effects for the type of support (categorical with four levels: Centor score, AI with Performance characteristics, AI with Explain by Feature Contribution, or AI with Explain by Examples), where Centor Score was set as the baseline group, and included control variables related to participants' characteristics such as years of experience and type of provider. We included a random effect for the participants' ID to account for the repeated measurements and potential individual variability. However, we dropped the random effect term when the variance components were estimated as zero, resulting in a simple linear regression with type of support as a fixed effect. In the tables reporting the results of regression models, the estimated value for each

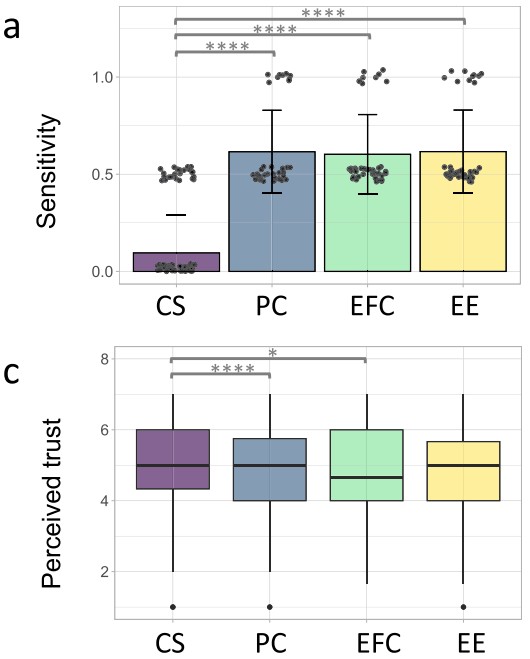

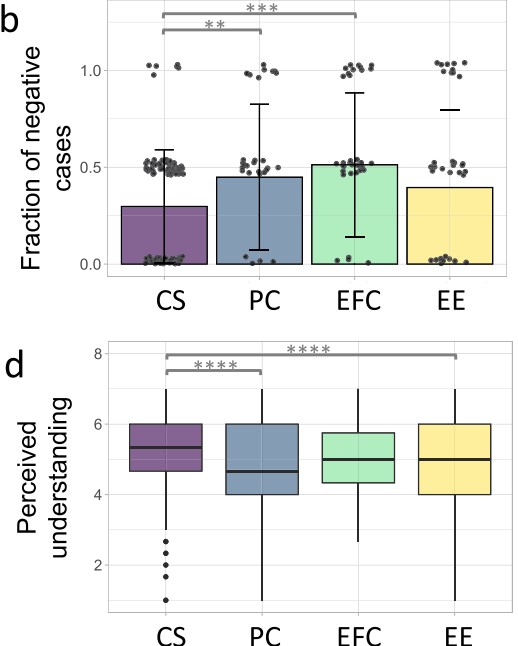

**Fig. 3 | Comparative analysis of CDSS types on outcomes and perceptions.** Plots of measures for different types of support information in the CDSS. CS corresponds to Centor Score, PC to Performance Characteristics, EFC to Explain by Feature Contribution, and EE to Explain by Examples. **a** Sensitivity ($N = 242$ observations), **b** fraction of negative cases that were suggested to order a confirmatory strep test ($N = 242$ observations), **c** perceived trust ($N = 968$ observations), and **d** perceived understanding of the CDSS ($N = 968$ observations). The error bars shown in the plots represent the standard deviation, and significant results are emphasized with gray lines. *$p \leq 0.05$, **$p \leq 0.01$, ***$p \leq 0.001$, ****$p \leq 0.0001$.

predictor indicates how much the mean outcome variable changes given a one-unit shift or change from the reference group to another level in the predictor while keeping the other predictors constant. The numbers reported for the intercept correspond to the value of the outcome variable when all predictor variables are zero. These tables also include the standard error, $t$-statistic, and $p$-value for each predictor. To analyze potential differences in continuous variables among the experimental groups with an AI-based CDSS, we used one-way analysis of variance (ANOVA) models. All post-hoc pairwise comparisons were conducted using Tukey's HSD test. For all the statistical tests reported below, $\alpha < 0.05$ was considered as a statistically significant effect.

## Results
We report the main findings in subsections that correspond to the main objectives of the study. From the valid data samples of 121 participants, the distribution over the experimental group was 39 in the performance characteristics, 39 in Explain by Feature Contribution, and 43 in Explain by Example.

### Diagnostic decisions are more accurate when participants work with an AI-based CDSS with explainable elements
We evaluated participants' ability to correctly diagnose all cases. More specifically, we measured the effect of the type of information presented in the CDSS on task performance metrics. We concentrated on presenting sensitivity and specificity results, recognizing the importance of quantifying the human–AI team's ability to correctly detect both ill patients out of those who do have the condition and healthy individuals. For a comprehensive evaluation, we complete the report with the metrics related to the proportions of positive and negative results that are true positive and true negatives. Table 2 summarizes the detailed outcomes of the regression for sensitivity and specificity measures. Participants reached significantly higher sensitivity values when they received diagnostic advice from an AI model with support information (all $p$-values $< 2.0 \times 10^{-16}$) compared to the use of the clinical prediction rule ($M = 0.10, SD = 0.20$). On average, sensitivity values were 0.62 ($SD = 0.21$), 0.60 ($SD = 0.21$), and 0.62 ($SD = 0.21$) with the support of performance characteristics, Explain by Feature Contribution, and Explain by Examples, respectively. Figure 3 presents the results for sensitivity using the different CDSS. Moreover, clinical experience between 11 and 15 years was generally associated with higher sensitivity than experience between 1 and 3 years ($p = 0.036$). Further comparisons using a one-way ANOVA did not reveal significant differences in sensitivity values between different types of explainable AI elements in the AI-based CDSS ($F(2, 118) = 0.05, p = 0.949$). As we compare specificity values obtained with the Centor score ($M = 0.75, SD = 0.26$), we observed significantly better ($p = 0.003$) results in Explain by Example ($M = 0.90, SD = 0.23$) and no differences in the other explanations groups (refer to Supplementary Fig. 2). On average, specificity values were 0.77 ($SD = 0.30$) and 0.65 ($SD = 0.35$) with the support of Performance Characteristics and Explain by Feature Contribution, respectively. In addition, a one-way ANOVA showed significant differences in specificity values across the AI explainable elements ($F(2, 118) = 6.87, p = 0.002, \eta_p^2 = 0.104$), but only between Explain by Example and Explain by Feature Contribution ($p = 0.001$). Similar to the sensitivity results, for positive and negative predictive value measures we found significant improvements in all the AI-based CDSS compared to the Centor Score. The complete descriptive statistics of positive and negative predictive values and the corresponding results of the linear models are summarized in Supplementary Tables 2 and 3, respectively.

### Higher agreement with diagnostic advice from an explainable AI-based CDSS than clinical prediction rule
We tested whether participants' agreement with diagnostic advice from the CDSS was affected by the type of information presented in the CDSS. Table 3 summarizes the results of the linear regression with the agreement metric as the outcome and the type of support as the independent variable.

**Table 3 | Agreement rate analysis across CDSS**

| Predictors | Estimate | Standard Error | $t$ | $p$-value |
|---|---|---|---|---|
| Intercept | 0.44 | 0.03 | 15.84 | $<2.0 \times 10^{-16}$ |
| PC | 0.27 | 0.03 | 9.26 | $<2.0 \times 10^{-16}$ |
| EFC | 0.21 | 0.03 | 7.22 | $1.2 \times 10^{-11}$ |
| EE | 0.34 | 0.03 | 12.13 | $<2.0 \times 10^{-16}$ |

Linear mixed model for agreement rate with the diagnostic advice. Centor score was the reference level. Random effects: $\sigma^2 = 0.001$, $N_{ID} = 121$, observations = 242, marginal $R^2 = 0.463$, conditional $R^2 = 0.476$.

**Table 4 | Analysis of confirmatory test recommendations for negative cases across CDSS**

| Predictors | Estimate | Standard error | $t$ | $p$-value |
|---|---|---|---|---|
| Intercept | 0.28 | 0.06 | 4.42 | $2.0 \times 10^{-5}$ |
| PC | 0.16 | 0.06 | 2.77 | 0.006 |
| EFC | 0.21 | 0.06 | 3.50 | $5.8 \times 10^{-4}$ |
| EE | 0.09 | 0.06 | 1.64 | 0.103 |

Linear mixed model for the fraction of negative strep cases that were recommended confirmatory testing. Centor score was the reference level. Random effects: $\sigma^2 = 0.024$, $N_{ID} = 121$, observations = 242, marginal $R^2 = 0.077$, conditional $R^2 = 0.268$.

Supplementary Fig. 2 provides additional visualizations. Participants agreed significantly more with the CDSS if the advice given to them was provided by an AI model with support information compared to the Centor score ($M = 0.42, SD = 0.14$). The effect was stronger in the Explain by Examples group (regression coefficient = 0.34, $p < 2.0 \times 10^{-16}$), followed by Performance Characteristics (regression coefficient = 0.27, $p < 2.0 \times 10^{-16}$) and Explain by Feature Contribution (regression coefficient = 0.21, $p = 1.2 \times 10^{-11}$). Furthermore, a one-way ANOVA revealed a significant main effect of the type of explanation presented in the AI-based CDSS on the agreement rate with predictions ($F(2, 118) = 5.46, p = 0.005, \eta_p^2 = 0.085$). Pairwise comparisons found that presenting predictions in Explain by Example ($M = 0.76, SD = 0.16$) resulted in significantly higher mean agreement than in Explain by Feature Contribution ($M = 0.63, SD = 0.20$), $p = 0.004$. However, there were no significant differences against the presentation of Performance Characteristics ($M = 0.69, SD = 0.17$).

### The explainable AI strategy in an AI-based CDSS is associated with certain follow-up actions
We analyzed whether the type of information presented in the user interface of the CDSS affected participants' follow-up actions. The results of the mixed-effects linear model are shown in Table 4. The average fraction of negative strep throat cases that were recommended confirmatory testing was higher when information related to Performance Characteristics ($M = 0.45, SD = 0.38$) and Explain by Feature Contribution ($M = 0.51, SD = 0.37$) supported the diagnostic prediction compared to the Centor score ($M = 0.30, SD = 0.29$), as it is shown in Fig. 3. More specifically, the effect was larger when the AI prediction was presented in the Explain by Feature contributions (regression coefficient = 0.21, $p = 5.8 \times 10^{-4}$) than using Performance Characteristics of the AI algorithm as support (regression coefficient = 0.16, $p = 0.006$). We did not find significant differences between the AI-based CDSS using Explain by Examples ($M = 0.40, SD = 0.40$) compared to the Centor score ($p = 0.103$). However, the differences between fractions of negative strep throat cases that were recommended confirmatory testing under different types of explanations in the AI-based CDSS were not significant ($F(2, 118) = 0.96, p = 0.387$).

**Table 5 | Analysis of perceived trust in the CDSS**

| Predictors | Estimate | Standard error | t | p-value |
|---|---|---|---|---|
| Intercept | 4.56 | 0.20 | 22.37 | $< 2.0 \times 10^{-16}$ |
| PC | −0.41 | 0.10 | −3.98 | $7.4 \times 10^{-5}$ |
| EFC | −0.24 | 0.10 | −2.36 | 0.019 |
| EE | −0.10 | 0.10 | −0.98 | 0.325 |

Linear mixed model for subjective trust ratings in the CDSS. Centor Score was the reference level. Random effects: $\sigma^2 = 0.657$, $N_{ID} = 121$, observations = 968, marginal $R^2 = 0.061$, conditional $R^2 = 0.452$.

**Table 6 | Analysis of perceived understanding of the CDSS**

| Predictors | Estimate | Standard error | t | p-value |
|---|---|---|---|---|
| Intercept | 5.18 | 0.19 | 27.21 | $< 2.0 \times 10^{-16}$ |
| PC | −0.58 | 0.09 | −6.63 | $5.7 \times 10^{-11}$ |
| EFC | −0.15 | 0.09 | −1.73 | 0.083 |
| EE | −0.39 | 0.08 | −4.66 | $3.7 \times 10^{-6}$ |

Linear mixed model for subjective understanding of the CDSS. Centor score was the reference level. Random effects: $\sigma^2 = 0.595$, $N_{ID} = 121$, observations = 968, marginal $R^2 = 0.070$, conditional $R^2 = 0.516$.

### Self-reported trust ratings decreased in the AI-based CDSS with feature-based explanations and displaying performance characteristics

We examined whether self-reported trust levels in the CDSS were affected by the type of information presented in the user interface. The results of the linear regression are shown in Table 5. Participants reported significantly lower trust levels, on average, when they were presented with information related to Performance Characteristics ($M = 4.65$, SD = 1.29, $p = 7.4 \times 10^{-5}$) and Explain by Feature Contribution ($M = 4.62$, SD = 1.26, $p = 0.019$) than when the interface presented the Centor Score ($M = 4.95$, SD = 1.32). More specifically, the estimated trust rating is 0.41 and 0.24 points lower when participants are presented with Performance Characteristics and in Explain by Feature Contribution, respectively, compared to the Centor Score. We did not find significant differences in trust ratings when participants used the AI with Explain by Examples ($M = 4.82$, SD = 1.10) compared to the Centor Score ($p = 0.325$). Figure 3 presents the results for perceived trust ratings. Moreover, considering the type of provider, nurse practitioners reported higher trust ratings than physicians ($p = 0.036$). When comparing subjective trust ratings from participants' interactions with the AI-based CDSS, a one-way ANOVA did not reveal significant differences ($F(2, 118) = 0.53$, $p = 0.590$).

### Perceived understanding of the CDSS was lower when performance characteristics were displayed and with example-based explanations

We compared participants' understanding of the CDSS when different information was presented to support the diagnostic advice. Table 6 summarizes the results of the mixed linear regression. Participants rated their understanding of the CDSS significantly lower when they were presented with Performance Characteristics ($M = 4.68$, SD = 1.41) and Explain by Examples ($M = 4.82$, SD = 1.17) compared to the Centor Score ($M = 5.20$, SD = 1.04), with $p$-values of $5.7 \times 10^{-11}$ and $3.7 \times 10^{-6}$, respectively. More specifically, the estimated understanding ratings of the CDSS are 0.58 and 0.39 points lower when participants were presented with its Performance Characteristics and Explain by Examples, respectively, compared to the Centor Score. We did not find significant differences in understanding ratings of the CDSS when participants used the AI-based CDSS with Explain by Feature Contribution ($M = 4.97$, SD = 0.91) compared to the Centor Score

($p = 0.083$). Figure 3 presents the results for perceived understanding ratings. When comparing subjective understanding ratings from participants' interactions with the AI-based CDSS, a one-way ANOVA did not reveal significant differences ($F(2, 118) = 0.77$, $p = 0.464$).

## Discussion

We conducted an empirical study to understand human interactions with an AI-based CDSS that relies on mobile-phone-acquired images for telehealth. As AI assistance is promising in the delivery of virtual health, it is imperative to evaluate how algorithmic advice affects providers or other stakeholders involved in decision-making in scenarios with considerably less opportunity to ensure decisions. We summarize the main findings related to the impact of explainable AI elements supporting strep throat screening on clinicians' decisions.

The results indicate that participants' ability to identify the presence or absence of strep throat cases is better when they receive diagnostic advice from an AI-based CDSS, regardless of the support information, compared to the Centor score. Specificity was only better in the Explain by Examples condition compared to the Centor Score. Compared to the metrics reported by studies that initially validated the Centor score[17], those that involve true positives counted in the computation have, on average, lower values when using the Centor score in our study. Nevertheless, we computed all metrics on a limited sample of four cases, including only two positive instances. Consequently, participants failing to identify these cases led to an absence of true positives, impacting the sensitivity and positive predictive value metrics, which were dominated by zeros. Even though state-of-the-art AI methods reach high performance on stand-alone operations, achieving a superior team performance outcome is a desired property in human–machine collaborations. Consequently, multiple studies have evidenced benefits in task performance when AI systems are used as decision support to clinical experts[60,61]. Depending on the task context, the type of model outputs might affect potential gains in human accuracy. For instance, for skin cancer diagnosis, presenting AI-based multiclass probabilities rather than the probability of malignancy or retrieving similar images with known diagnoses resulted in better team accuracy than decisions without any support[5]. The superior performance demonstrated by clinicians' interactions with AI-based CDSS is promising for smartphone-based strep throat prediction and contributes to safe deployment strategies for these systems[62].

By comparing participants' final diagnosis with the provided diagnostic advice, we quantified the level of agreement under different types of support to identify strep throat cases. We found that agreement with diagnostic advice increases when it comes from an AI model compared to the Centor Score. Previous studies on reliance on AI advice have observed that people are willing to follow AI's advice, which might result in undesired outcomes if the algorithms err[63,64]. Participants' task knowledge can also affect their willingness to adopt AI's advice[65,66]. In the context of remote strep throat screening with no access to laboratory tests or healthcare professionals operating without any form of medical tests, clinicians only have access to visual data of the throat and the variability in the presentation of strep pharyngitis makes it challenging to identify. Even though the predictions of the AI and the Centor score always suggested a correct diagnosis, the way in which advice is presented can affect users' decisions[67]. While the diagnostic advice was presented explicitly in all the AI-based CDSS (POSITIVE/NEGATIVE), the suggestion in the Centor score included the likelihood (overall low probability) of strep pharyngitis. Moreover, we observed a significantly higher agreement with AI diagnostic advice when presenting Explain by Examples information than Explain by Feature Contribution. Even though our results suggest the advantages of supporting comparisons to diagnose, presenting diagnoses in similar cases might become less important as clinicians get familiar with the decision support system[32].

In addition, our evaluation of subjective measures suggests higher satisfaction with the standard clinical prediction rule compared to some of the AI-based CDSS. Compared to the Centor Score, we observed lower self-reported trust ratings on diagnostic advice in the Explain by Feature

Contribution and Performance Characteristics of the AI model. We attribute this difference to the fact that the Centor score is a practice guideline endorsed by many prestigious organizations, such as the American College of Physicians, in addition to being a commonly accepted tool and fairly well-known by clinicians. Another factor shaping trust is clinicians' risk tolerance with uncertainty, as presented in the Centor Score. Pairing AI predictions with confidence levels that communicate uncertainty is currently a subject of study in the field of human–AI interaction, including the complex interplay with explanations[68]. Evaluations involving radiologists, for instance, suggest that trust in an algorithm for breast lesion classification was generated primarily by the certainty of the prediction, though these findings were drawn from a study with a small sample size[44]. Even though participants perceived lower trust in the AI's diagnostic advice, this finding is not consistent with participants being more likely to agree with advice from the AI-based CDSS. Therefore, the AI's advice can influence participants' decisions, but they consciously perceive they do not trust such advice. This mismatch between what participants perceived and reported and how they behaved has been identified before in empirical studies[69,70]. Meanwhile, no significant differences in perceived trust were found when the diagnostic advice had the Explain by Examples support. Participants reported a lower understanding of the AI-based CDSS with Performance Characteristics and Explain by Examples compared to the Centor Score. The direct presentation of performance outcomes has limited abilities to inform about the inner mechanisms of the CDSS, and might still be perceived as a black box model. Even though example-based explanations can provide an idea of how the AI model operates, extracting an appropriate mental model with a few cases is limited. Alternatively, Feature Contributions present visual examinations that clinicians are familiar with and more closely resemble the use of a patient's attributes in the Centor Score.

The higher levels of trust using the Centor Score were further reflected in a reduced number of cases in which participants recommended confirmatory testing for negative cases of strep throat. In particular, participants were more likely to recommend confirmatory testing for cases with no strep throat when the support information included Performance Characteristics and Explain by Feature Contribution, which is consistent with the lower self-reported trust ratings on these experimental groups. Unlike the Centor score, AI-based clinical decision support is not a routine part of the clinical workflow and remains largely outside the historical experience of providers. In a risk-averse clinical environment, this unfamiliarity with AI tools may result in clinicians' skepticism toward AI's advice. Instead, they would rather rely on a known/trusted tool (Centor) or their own clinical judgment for identifying patients whose risk for streptococcal infection is so low that microbiological testing or antibiotic treatment is unnecessary and the clinical criteria suffice[71]. Consistently, we did not observe this effect when Explain by Examples was used to support the AI's diagnostic advice. However, we note that the average fraction of negative strep throat cases that were recommended confirmatory testing was not minuscule (greater than 30%) under the Centor Score and use of AI with Explain by Examples. The reason clinicians recommend more follow-up for negative tests might be because of low-risk tolerance when treating patients; they are not willing to risk a misdiagnosis, regardless of the CDSS format. Evaluating follow-up actions is crucial for telehealth as this determines the feasibility of incorporating AI support to screen for disease remotely, i.e., whether providers can accurately identify patients who can be treated remotely while being confident in their diagnoses. For instance, in addition to disease category prediction, an evaluation of management decisions revealed that dermatologists with AI support switched from more invasive treatments to simple monitoring for benign lesions, showing a potential to decrease the number of unwarranted interventions and costs[5].

The current study has some limitations. First, it exhibits the common limitations of survey studies, including convenience sampling that might not reflect the true target population. Likewise, to maintain a reasonable study duration, in this pilot, we only considered cases where the diagnostic advice was correct. However, we acknowledge the different implications of agreeing with correct or incorrect advice, which is expected when testing a

real rather than a mock-up algorithm, and how participants' performance can be impaired. We recommend future works to explore how different types of AI errors and their presentation affect trust and adoption of advice in the presence of explanations that promote trust and understanding. The complexity of the study design can increase when revealing the performance indicators of the algorithm, which is why we established predetermined values considered good enough for clinical acceptance in the PC group. However, it remains uncertain whether the findings would be consistent across a different range of metric values. Second, by focusing on a carefully chosen subset of explanation techniques, we aimed to balance the depth of investigation with practical constraints without overloading participants or drastically increasing the cost of the study, considering that the current experimental setup already included a large sample size of stakeholders. However, we acknowledge that more techniques are worth exploring to address the limitations identified in this study. Third, the CDSS presented diagnostic advice directly during the study, rather than allowing participants to work on the task on their own first, as this might affect whether they incorporate the diagnostic advice and can capture their behavioral intention to use the AI. Lastly, the limited data samples per participant and, therefore, the variability in the control and experimental group might have affected the capabilities of the statistical models used to fit the data. Therefore, we followed a conservative interpretation of the results and the derived claims.

In conclusion, AI provides opportunities to transform telehealth and the study of human interactions with AI-based CDSS is crucial for its adoption to new tasks. Our main finding is that different types of explainable AI elements supporting strep throat predictions generated by an AI might have influenced clinicians to follow AI's advice more often. However, clinicians were not fully satisfied in terms of trust or understanding of the CDSS when they received AI assistance, as illustrated by the Explain by Examples being more trustworthy but more difficult to comprehend. These considerations can potentially apply to similar types of studies or scenarios, where clinicians do not have the opportunity to reassure decisions. Different human factors should be considered when designing explanations or presenting support information to achieve certain goals with end users of AI systems, which include but are not limited to improving decisions, supporting transparency, and moderating trust. Improving one aspect might compromise others, e.g., better understandability at the cost of increasing cognitive load, or more generally, the costs and benefits of engaging with an AI explanation. Overall, efforts should be distributed to achieve high performance while accounting for human behavior in practice, especially in a healthcare context where guiding clinicians' decisions has a direct impact on patients' well-being.

## Data availability
Patient data used in this study cannot be made publicly available to protect the privacy and safety of research participants. The de-identified patient data can be shared from the corresponding author, pending approval from the Institutional Review Board, and may require the execution of a data use agreement between the party requesting to use the data and Johns Hopkins University. The survey data that supports the findings of this study, including the numerical data used to plot Fig. 3 (source data) is available in Supplementary Data 1.

## Code availability
The free programming language R (4.2.2) was used to perform all statistical analyses. The code used to generate the results, as described in the manuscript, is available in Supplementary Data 1.

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

## Acknowledgements

Ashtine Rodriguez for creating the prototypes for the study. This work was supported by the Bisciotti Foundation Translational Fund at Johns Hopkins University and Johns Hopkins University internal funds.

## Author contributions

Conceptualization: T.C., M.U., C.G., A.Z. Methodology: formal analysis, investigation, and visualization: C.G., T.C., A.Z., B-L.S. Funding acquisition: T.C. Supervision: T.C., M.U. Writing—original draft: C.G. Writing—review & editing: T.C., M.U., C.G.

## Competing interests
The authors declare the following competing interests: contributing author, T.C., has a co-inventor status on the patent of the technology referenced, and ownership in a company that is commercializing the technology described in the article. Currently, the product described is not promoted nor commercially available. Contributing author, M.U., is a co-inventor of the

patent as well. Authors C.G., B.-L.S., and A.Z. declare no competing interests.
