## [Peer Review File · Communications Medicine]

Reviewers' comments:

Reviewer #1 (Remarks to the Author):

Thank you for inviting me to review this highly topical and useful study comparing different types of AI-based clinical decision support on clinicians' accuracy and trust (in the AI CDSS) for diagnosing strep pharyngitis in a simulated telehealth context.

I have enjoyed reading this manuscript, it is well written and I expect will be of interest to a range of audiences including clinicians, developers, and potentially healthcare managers and policy makers (irrespective of whether or not they are familiar with AI). Many aspects of the study was well designed and conducted and my specific comments below are more about clarity than any fundamental concerns with the validity of the findings and conclusions drawn. However, I would urge the researchers to consider developing a more in-depth discussion to help better explain their findings in the context of real world clinical practice, human behaviour and specific human factors aspects they repeatedly refer to.

Specific comments:

- Line 119 "a mixed design" is not really a study design. Suggest making this clearer e.g. 3-arm comparison and cross-over study (if that is the case?), or perhaps a diagram illustrating participant allocation would be helpful here.
- If I understand this correctly, line 137 "the ideation process departed from the current clinical practice" simply means the team did not use the exact same parameters in the AI experiments as would be presented by Centor? If so, I would suggest this be made more explicit to minimise risk of this being perceived as a negative point ie. Not taking into consideration current clinical practice. Instead perhaps rephrase such as "the team developed different parameters in the AI-based CDSS prototype designs that built on current clinical practice" (if that was the case).
- Line 142, please explain what 'held-out data' is.
- Figure 1 is very useful and actually much clearer to understand the different arms
- I am quite surprised by the sensitivity values across all arms, particularly in the Centor score arm. How does this compare to other studies in the literature?
- Section 3.1 seems to be missing data on specificity
- Lines 265 onwards describes diagnostic recommendations but the types of diagnostic recommendation options were not previously described in the methods?
- This section seems limited despite the interesting findings. I would expect more discussion around potential and/or reported reasons in the literature for why clinicians had more trust in the Centor score despite it being less accurate e.g. repeated use and familiarity with the tool? Other human factors' considerations e.g use of binary language (negative or positive for strep pharyngitis) in AI suggestions vs % likelihood for Centor score – clinicians will be familiar with a degree of diagnostic uncertainty without a definitive diagnostic test

Reviewer #2 (Remarks to the Author):

The paper presents a study involving a panel of 121 health professionals, recording their responses as part of a randomised trial regarding the remote diagnosis of streptococcal pharyngitis. The goal is to compare four methods for assisting the experts' diagnoses, one based on a "traditional" score

(Centor score), and three based on AI predictions. Each of the three methods supports its advice using a different set of explanations.

The main finding and claim in the paper is that participants tend to agree more with AI-based recommendations than with the Centor score, but at the same time when asked to respond to a questionnaire, they report lower trust in those systems.

Novelty:

While there is some novelty here, it is hard to imagine this is the first study on this topic, given the increasing momentum around XAI and its role in support of clinical practice. In this perspective, the paper does not offer much in terms of related work, in a way that may help appreciate the novelty of either the methods, or of the results.

Nevertheless, there is merit in the paper as it attempts to quantify expert perception of new type of advice that is available to them in support of remote diagnosis through AI prediction tools, and uses standard statistical methods (mainly mixed-effects linear models) to analyse the results.

In my view the major point of contention in the study is that this is actually a "what if" scenario, where mock ups of AI systems are used in lieu of the real thing. This begs the question of whether this study design, and especially the manual design of the explanations themselves (if I understand correctly), is enough to prove the point and support the claim. In each of the three cases, the design of the specific explanations presented to the clinicians should be justified. For example for PC, how were the accuracy measures identified? clearly those figures affect the trust levels, and one may argue that the conclusions from the experiment should be qualified with "trust in PC as an explanatory method *at given levels of stated performance*". This however leads to a much narrower conclusion than what is claimed here, as arguably perception is driven, at least in part, by those specific figures.

Similarly, there are well-documented methods for assessing feature contribution (EFC) in ML practice (see eg Shapley values), and unless the relative contributions presented to the participants in the mock up reflect what the models can actually do, this diminishes the socio-scientific significance of the findings.

If we accept that the exact presentation of the explanations has an influence on trust and perception of the AI advice, that arguably this type of study should cover a much larger space of options than the authors have explored.

A second, perhaps less critical "shortcut" that may limit the strength of the results, is the assumption that the AI is, in fact, infallible. In other words, that the AI predictions are always correct, with no False Positives or False Negatives.

While this assumption does not invalidate the study (assuming, of course, that participants do not have such information), it is completely unrealistic, and again weakens the conclusions by disregarding all the cases where the clinician's diagnosis is actually superior to that of the AI, i.e., when there is disagreement *and* the clinician is correct. I feel that for the study to provide realistic and usable conclusions, this strong assumption should be relaxed, and the possibility of FP/FN should be included.

In summary, there is merit in the study, but its impact would be greater if the assumptions used for the mock-up are either relaxed, or fully justified, i.e., by undertaking a more comprehensive analysis of each of the explanation options -- and possibly using real AI-based CDSS.

Reproducibility: I don't see enough detail in the paper or appendices to reproduce the experiments.

We thank both reviewers and editors for their constructive criticism. We restate the primary contribution of our paper for easier contextualization of our responses. Our main contribution is the ideation and evaluation of an Artificial Intelligence (AI)-based decision support tool for the remote and asynchronous screening of infectious diseases. Specifically, this study evaluates physicians' understanding and trust of an AI-based clinical decision support system in the context of medical decision making for pharyngitis, based on smartphone images of the throat acquired by the patients. In what follows, we respond to all concerns raised during review and detail the changes that we made to improve our manuscript.

The major changes of the manuscript are as follows:

- To clearly justify the design choices of the explainability elements for the AI predictions of strep throat, we provided rationales that support how we crafted the prototypes that illustrate each technique.
- To better support the definition of the study design, we commented on the rationale for including only accurate predictions and the use of a mock-up AI model and acknowledged the limitations associated with these choices, ensuring an understanding of their impact on our findings.
- To better situate the results of our user study, we complemented the discussion with interpretations of our findings considering current clinical practice.

In the remainder of the response letter, we use black bold font to state reviewers' comments and purple bold font to highlight the major concerns. We use black font for our answers. We also mark changes in the manuscript with blue color and attached text fragments with changes in the letter.

Reviewer 1:

Q: I have enjoyed reading this manuscript, it is well written and I expect will be of interest to a range of audiences including clinicians, developers, and potentially healthcare managers and policy makers (irrespective of whether or not they are familiar with AI). Many aspects of the study were well designed and conducted and my specific comments below are more about clarity than any fundamental concerns with the validity of the findings and conclusions drawn. However, I would urge the researchers to consider developing a more in-depth discussion to help better explain their findings in the context of real world clinical practice, human behaviour and specific human factors aspects they repeatedly refer to.

A: We agree that elaborating more on discussion of our results will better highlight the insights from our user study. The human factors we considered included effectiveness, willingness to use, trust, and understanding of the novel AI-based CDSS. In terms of effectiveness to correctly identify strep cases, the lower performance in the Centor score group can be explained by the fact that this rule is used in clinical practice to guide whether a diagnostic test is indicated and has a low positive predictive value. Consistently, our results show low PPV and sensitivity using the Centor score.

Differences in perceived trust can be attributed to familiarity with the CDSS (the next comment addresses more on trust). Likewise, perceived understanding of the CDSS can be affected by familiarity with the tool itself and comprehension of the mechanisms that guide its functioning, which may be more limited for the explanation with performance and examples.

These details have been incorporated in the corresponding paragraphs in the discussion and presented in more detail in the next question.

Q: This section seems limited despite the interesting findings. I would expect more discussion around potential and/or reported reasons in the literature for why clinicians had more trust in the Centor score despite it being less accurate e.g. repeated use and familiarity with the tool? Other human factors' considerations e.g use of binary language (negative or positive for strep pharyngitis) in AI suggestions vs % likelihood for Centor score – clinicians will be familiar with a degree of diagnostic uncertainty without a definitive diagnostic test

A: We agree that including potential reasons for observing higher trust levels in the Centor score will enrich the discussion and thank the reviewer for the suggestions; both reasons are relevant and could explain differences in trust findings. Regarding the use of language, we mentioned how the binary language vs presenting likelihood values may have affected agreement with recommendations. As suggested by the reviewer, clinicians' familiarity with uncertainty in diagnosis could have affected their reported trust. We have expanded the discussion of trust differences (line 409):

“We attribute this difference to the fact that the Centor score is a practice guideline endorsed by many prestigious organizations, such as the American College of Physicians, in addition to being a commonly accepted tool and fairly well known by clinicians. Another factor shaping trust is clinicians' risk tolerance with uncertainty, as presented in the Centor Score. Pairing AI predictions with confidence levels that communicate uncertainty is currently a subject of study in the field of human-AI interaction, including the complex interplay with explanations [67]. Evaluations involving radiologists, for instance, suggest that trust in an algorithm for breast lesion classification was generated primarily by the certainty of the prediction, though these findings were drawn from a study with a small sample size [43].”

And line 436:

“Unlike the Centor score, AI-based clinical decision support is not a routine part of the clinical workflow and remains largely outside the historical experience of providers. In a risk-averse clinical environment, this unfamiliarity with AI tools may result in clinicians' skepticism toward AI's advice. Instead, they would rather rely on a known/trusted tool (Centor) or their own clinical judgment for identifying patients whose risk for streptococcal infection is so low that microbiological testing or antibiotic treatment is unnecessary and the clinical criteria suffice [70].”

Q: Line 119 “a mixed design” is not really a study design. Suggest making this clearer e.g. 3-arm comparison and cross-over study (if that is the case?), or perhaps a diagram illustrating participant allocation would be helpful here.

A: We agree that the terminology used should be changed to clearly communicate the study design used and have included a diagram to illustrate participant allocation. Following the suggestion, we modified the description of our study (line 139) and added a diagram (new Figure 1):

“We defined a randomized three-arm comparison experiment, where each arm corresponds to an explainable AI element. Participants started by completing the diagnostic task using their clinical judgment supported by the Modified Centor score criteria (hereafter referred to as Centor Score). Then, they were randomly assigned to one of the three arms in which AI predictions were augmented with information that can be understandable to the end users, as in a between subjects design.”

Q: Line 137 “the ideation process departed from the current clinical practice” simply means the team did not use the exact same parameters in the AI experiments as would be presented by Centor? If so, I would suggest this be made more explicit to minimise risk of this being perceived as a negative point ie. Not taking into consideration current clinical practice. Instead perhaps rephrase such as “the team developed different parameters in the AI-based CDSS prototype designs that built on current clinical practice” (if that was the case).

A: We agree that the current phrasing of that sentence can communicate a negative point, especially since we did consider current clinical practices and the medical expertise in our team to define the transparency elements. We modified the sentence as follows in line 157:

“Within our multidisciplinary team of computer scientists, clinicians, and user interface designers, the ideation process was based on current clinical practice and transparency techniques previously used in medical tasks [30] to formulate the following transparency elements to support AI predictions:”

Q: Line 142, please explain what ‘held-out data’ is.

A: In the description of the performance characteristics group, we use the terminology ‘held-out’ data to refer to data not used during training. This is a standard in the evaluation of machine learning models to assess how the model will perform on unseen data, providing an indicator of the model’s true performance. We modify the description of the Performance Characteristics in the Methods line 162 as follows:

“Evaluating the AI model on held-out data (i.e. data not used for model training) using task-related metrics is part of the algorithm pipeline to ensure that the model performs well not just on the data it was trained on, but also on new, unseen data. Such indicatives of the model's

true performance can be used to communicate the stated performance, usually on a test set, before deployment where novel instances may be encountered [53].”

Q: I am quite surprised by the sensitivity values across all arms, particularly in the Centor score arm. How does this compare to other studies in the literature?

A: We have reviewed the distribution of sensitivity values across all arms. In the Centor Score group, sensitivity values are dominated by zero, meaning that the number of true positives is zero. Additionally, we note that there were only two positive cases in each experimental arm, which may limit the variability in the potential outcomes of the prediction task, i.e., the number of true positives is either zero, one, or two. More details about the selection of vignette cases is given in the Procedure paragraph and we explained differences on the metrics that involve true positives in the discussion line 375 as follows:

“Compared to the metrics reported by studies that initially validated the Centor score [17], those that involve true positives counted in the computation have on average lower values when using the Centor score in our study. Nevertheless, we computed all metrics on a limited sample of four cases, including only two positive instances. Consequently, participants failing to identify these cases led to an absence of true positives, impacting the sensitivity and positive predictive value metrics, which were dominated by zeros.”

Q: Section 3.1 seems to be missing data on specificity

A: For a more comprehensive presentation of our results, we present the findings for both sensitivity and specificity measures in the first paragraph at Section 3.1 and Table 1, and mention that other performance metrics are provided in the Appendix section D. We have included these additional metrics in the appendix to maintain the focus and readability of the main text. We have improved the clarity for the reader to find these results:

Line 278:

“More specifically, we measured the effect of the type of information presented in the CDSS on task performance metrics. We concentrated on presenting sensitivity and specificity results, recognizing the importance of quantifying the human-AI team ability to correctly detect ill patients out of those who do have the condition. For a comprehensive evaluation, we complete the report with the metrics related to the proportions of positive and negative results that are true positive and true negatives in the Appendix.”

Line 294:

“As we compare specificity values obtained with the Centor score ($M=0.75$, $SD=0.26$), we observed significantly better ($p=.003$) results in Explain by Example ($M=0.90$, $SD=0.23$) and no differences in the other explanations groups. On average, specificity values were 0.77 ($SD=0.30$) and 0.65 ($SD=0.35$) with the support of Performance Characteristics and Explain by Feature Contribution, respectively. In addition, a one-way ANOVA showed significant differences

on specificity values across the AI explainable elements ($F(2, 118)=6.87$, $p=.002$, $\eta^2=.104$), but only between Explain by Example and Explain by Feature Contribution ($p=.001$)."

Q: Lines 265 onwards describes diagnostic recommendations but the types of diagnostic recommendation options were not previously described in the methods?

A: To improve clarity and avoid confusion, we would like to differentiate what we meant by the recommendations provided by the CDSS and those that participants were asked to make for each clinical vignette. For consistency, we have adopted the terminology diagnostic advice to refer to the outcomes displayed by the decision support system, which were positive/negative predictions in the AI models. The definition is in line 181:

"The AI-based diagnostic advice corresponded to a prediction of either a positive or negative diagnosis for strep throat."

The follow-up actions of the telehealth visit were specified in the Methods line 219 and use the term follow-up actions in the Results and Discussion sections as well.

"Participants had to determine whether each case was positive or negative for strep throat and choose among three potential follow-up actions, as shown in Fig. 3. These included advising supporting care, a treatment with antibiotics, or a referral for a strep test."

Reviewer 2:

Q: While there is some novelty here, it is hard to imagine this is the first study on this topic, given the increasing momentum around XAI and its role in support of clinical practice. In this perspective, the paper does not offer much in terms of related work, in a way that may help appreciate the novelty of either the methods, or of the results.

A: We agree that situating our work in the existing literature of XAI is crucial to better appreciate our contribution. The novelty of this work lies in evaluating the promising opportunity to support remote and asynchronous diagnosis combining powerful AI algorithms and alternative data sources provided by technologies that are more accessible. A crucial dimension for the adoption of these novel decision support tools includes understanding how providers will use and interpret such advice, which is currently quantified in our study. In the introduction, we motivate the need for evaluating promising AI tools for the implementation of telehealth in line 72:

"Under this assistance scenario, both the clinician and AI algorithm review an image, the main source for the diagnostic decision. Remote review of smartphone images is not a standard of care currently, nor is it consistently performed with mobile or remote monitoring devices beyond conventional clinical settings. In particular, clinicians have no unified practice in evaluating pharyngitis through telehealth, providing an opportunity to study interpretations and

effectiveness for such an AI CDSS to be integrated and expand diagnostic capacities outside the clinic.”

Besides, we expanded the presentation of related works in the introduction and provided a more clear distinction when we contrast our study with previous empirical evaluation of AI support in healthcare settings.

Line 97:

“In domains such as breast imaging and dermatology, where deep learning models have been well-established and extensively studied [5, 32], there exists a notable interest in empirical evaluations of explainability techniques that can be directly incorporated into existing models.”

Line 102:

“In parallel, certain studies have chosen to introduce explainability features into interfaces all at once [41, 42, 43], which can lead to convoluted user experiences, thereby hindering the capacity to evaluate their efficacy or pinpoint which components are genuinely beneficial. In less explored domains, there is a surge in innovating explanation methodologies utilizing mock-ups to pilot novel concepts, as evidenced in tasks such as antidepressant [37] and chemotherapy prescriptions [44], or automated biosignal analysis [45, 46].”

Line 115:

“We must meticulously conceptualize AI's role in the remote screening of infectious diseases from non-conventional image sources, exploring essential assistance elements to foster successful collaborations with the envisioned stakeholders.”

Q: In my view the major point of contention in the study is that this is actually a "what if" scenario, where **mock ups of AI systems are used in lieu of the real thing. This begs the question of whether this study design, and especially the **manual design of the explanations themselves** (if I understand correctly), is enough to prove the point and support the claim.**

A: We agree with the importance of justifying design choices in a mock-up AI system and explain why we did not use a real AI model for this study. Building a model with an explainability mechanism that will not likely achieve certain human factor goals may result in misspent efforts and resources in developing such complex models. Therefore, we followed a human-centered design approach where we first approached end users and understood the context to ideate prototypes that could be rapidly tested with the target audience. As correctly understood by the reviewer, we developed high-fidelity prototypes that illustrate the functionality of the AI-based CDSS and the transparency elements, which were manually designed. In this way, future iterations and refinement of design choices can be completed without retraining or re-formulating the AI models. We added the following justification for following a human-centered approach with a mock-up AI system in the introduction line 128:

“In the formulation of such an AI system, we implemented core practices of Human-Computer Interaction (HCI) design, approaching end users and developing a mock-up AI system as an early-stage interactive AI prototype [49, 21]. This approach allows us to rapidly explore design opportunities and probe user behaviors, without compromising resources in the implementation of a functioning AI system that may not achieve certain human factors goals.”

We hope that our study informs the development and implementation of algorithms that can afford trust and understanding beyond our particular application, but similar scenarios where experts cannot reassure their decisions. We mention this in the concluding paragraph line 474:

“Our main finding is that different types of explainable AI elements supporting strep throat predictions generated by an AI might have influenced clinicians to follow AI's advice more often. However, clinicians were not fully satisfied in terms of trust or understanding of the CDSS when they received AI assistance, as illustrated with the Explain by Examples being more trustworthy but more difficult to comprehend. These considerations can potentially apply to similar types of studies or scenarios, where clinicians do not have the opportunity to reassure decisions.”

Q: In each of the three cases, the design of the specific explanations presented to the clinicians should be justified. For example for PC, how were the accuracy measures identified? clearly those figures affect the trust levels, and one may argue that the conclusions from the experiment should be qualified with "trust in PC as an explanatory method *at given levels of stated performance*". This however leads to a much narrower conclusion than what is claimed here, as arguably perception is driven, at least in part, by those specific figures

A: We agree with this observation as previous studies have considered the interpretability of performance indicators for people to evaluate the overall reliability of an AI model [1, 2] . The complexity of studying the effect of presenting performance metrics increases if one considers more factors, such as the stated versus observed performance across a set of AI predictions. Although the AI model's accuracy may affect trust and other behavior measurements, we defined the performance metrics to be used in the study such that they would meet clinically acceptable performance, as determined by preceding interviews with clinicians (refer to line 186). In our conclusion, we clarify that we report performance indicators of models that are good enough to be used in clinical practices. We acknowledge potential limitations on the generalizability of this result in the limitations paragraph in the discussion line 459 as follows:

“The complexity of the study design can increase when revealing the performance indicators of the algorithm, which is why we established predetermined values considered good enough for clinical acceptance in the PC group. However, it remains uncertain whether the findings would be consistent across a different range of metric values.”

[1] Yin, Ming, Jennifer Wortman Vaughan, and Hanna Wallach. "Understanding the effect of accuracy on trust in machine learning models." In *Proceedings of the 2019 chi conference on human factors in computing systems*, pp. 1-12. 2019.

[2] Rechkemmer, Amy, and Ming Yin. "When confidence meets accuracy: Exploring the effects of multiple performance indicators on trust in machine learning models." In *Proceedings of the 2022 chi conference on human factors in computing systems*, pp. 1-14. 2022.

Q: Similarly, there are well-documented methods for assessing feature contribution (EFC) in ML practice (see eg Shapley values), and unless the relative contributions presented to the participants in the mock up reflect what the models can actually do, this diminishes the socio-scientific significance of the findings.

A: We reviewed existing explainability techniques, including explanations that are computed post-hoc such as Shapley Additive explanations [1]. In our formulation of Explain by Feature Contribution, we used bar plots to illustrate the relative contribution of attributes that favor a positive or negative strep prediction informed by discussions with experts that constantly screen patients with strep throat. In practice, experts can provide annotations of these attributes using the images of the throat; therefore, their influence can be computed in the model's predictions for a particular instance, as in a local explanation. As mentioned by the reviewer, Shapley values would be one possible way to compute these weights for a trained classifier that predicts positive or negative strep cases. We include these practical considerations in the description of Explain by Feature Contribution in the Methods line 172:

"In practice, by annotating these signs on throat images, their influence can be computed in a trained classifier that predicts positive or negative strep cases, using tools like Shapley values for example."

[1] Lundberg, Scott M., and Su-In Lee. "A unified approach to interpreting model predictions." *Advances in neural information processing systems* 30 (2017).

Q: If we accept that the exact presentation of the explanations has an influence on trust and perception of the AI advice, that arguably this type of study should cover a much larger space of options than the authors have explored.

A: Determining what constitutes a good explanation for a specific audience within a novel application, as it is the remote prediction of strep throat, continues to be an unresolved question, especially within the broad context of design options. To avoid over-investing resources in an idea that may not work, we emphasize the need to start by mocking-up prototypes to explore multiple design possibilities, and test them to assess the human consequences of a design and iteratively improving on it. Furthermore, in the framework of user interface design, we acknowledge the importance and influence of the presentation style of explanations, previously reported by the literature on human-AI interaction [1, 2]. For the selection of explanation types, we reviewed and considered multiple explainability techniques. However, as pointed out by recent surveys [3], an explainability technique developed in a computer vision context may not be suitable or clinically meaningful in a specific domain. We selected three presentation types

that have the potential of being understandable by end users from our conversations and ideation process with experts. Despite our effort to include a variety of information elements that could potentially be understandable by the end user, we agree that more explainability elements should have been covered. However, this adds complexity to the study design, requiring larger samples or increasing participants' burden, which is costly. We added this to the limitations paragraph in the discussion line 462:

“Second, by focusing on a carefully chosen subset of explanation techniques, we aimed to balance the depth of investigation with practical constraints without overloading participants or drastically increasing the cost of the study, considering that the current experimental setup already included a large sample size of stakeholders. However, we acknowledge that more techniques are worth exploring to address the limitations identified in this study.”

In the conclusion, we encourage the ideation of novel transparency elements that can address the issues raised by our study, lower trust levels for example, and their thorough evaluation with end users. Actually, we have modified the part of the manuscript's title related to trust findings from “Yet Still Struggles to Gain Clinicians' Trust” to “Highlights Opportunities to Enhance Clinicians' Trust”.

[1] Cheng, Hao-Fei, Ruotong Wang, Zheng Zhang, Fiona O'connell, Terrance Gray, F. Maxwell Harper, and Haiyi Zhu. "Explaining decision-making algorithms through UI: Strategies to help non-expert stakeholders." In *Proceedings of the 2019 CHI conference on human factors in computing systems*, pp. 1-12. 2019.

[2] Wang, Xinru, and Ming Yin. "Are explanations helpful? a comparative study of the effects of explanations in ai-assisted decision-making." In *26th international conference on intelligent user interfaces*, pp. 318-328. 2021.

[3] Chen, Haomin, Catalina Gomez, Chien-Ming Huang, and Mathias Unberath. "Explainable medical imaging AI needs human-centered design: guidelines and evidence from a systematic review." *NPJ digital medicine* 5, no. 1 (2022): 156.

Q: A second, perhaps less critical "shortcut" that may limit the strength of the results, is the assumption that the AI is, in fact, infallible. In other words, that the AI predictions are always correct, with no False Positives or False Negatives. While this assumption does not invalidate the study (assuming, of course, that participants do not have such information), it is completely unrealistic, and again weakens the conclusions by disregarding all the cases where the clinician's diagnosis is actually superior to that of the AI, i.e., when there is disagreement *and* the clinician is correct

A: We acknowledge this limitation (line 395). Considering other factors, such as the type of AI prediction errors, adds complexity to the study design and requires further vignettes to include a representative sample of each type of AI prediction. In the current study, we did not evaluate the types of errors that the AI will likely make and how participants would perceive and respond to those cases with incorrect outputs of the AI. To confirm, participants were not aware of this and we did not communicate the AI's performance before the study, as this was one of the experimental manipulations. We mention this explicitly in the Methods line 199 to avoid confusion:

“Given the limited number of cases, the AI predictions were always correct, i.e., positive for confirmed cases with a throat culture and negative otherwise. This translates to an observed accuracy of 100 % during the study; however, participants were not aware of this and we did not communicate performance-related information given that one of our experimental manipulations involves the interpretation of performance characteristics.”

Q: I don't see enough detail in the paper or appendices to reproduce the experiments.

A: The reporting summary that is added to the main manuscript includes details of the study design to improve reproducibility. However, we agree that a complete set of visualizations and metadata for all the vignettes is missing (can be shared upon request). The data analysis code with the models used to support our findings can be shared after publication of this work.

REVIEWERS' COMMENTS:

Reviewer #2 (Remarks to the Author):

this is going to be a very short review for the revision that was submitted.

I believe you have addressed the main points I raised. Of course the main limitations, which you seem to agree with, remain, but those are now acknowledged and the points of contention have been clarified.

The HCI approach is acceptable, though I still believe that, for example, assuming infallible AI is a bit too much of a simplification. I am sure there will be scope for further work that will relax this and other assumptions.

Reviewer #3 (Remarks to the Author):

This paper is significantly topical considering the relatively recent outbreaks in Group A strep infections and with the increased workload placed on primary care, there is an emphasis on the role of telemedicine in triaging patients remotely.

This paper highlights the need to reinforce the workforces trust in AI as decision support tools. It also suggests a way in which AI can be used to guide antibiotic prescribing to ensure limitation of risks of complications of sepsis but also antimicrobial resistance.

In thinking about the practicality of this model in terms of diagnosis of strep A infections without the need for face to face consultation, I think we need to consider the practicalities of not only diagnosis but also risk management and identifying sick patients. I feel as a tool to fully manage a patient remotely in a safe manner, it would need to be considered how risk is managed i.e. how observations could be measured.

I have noted that the writers have responded to prior feedback from the other reviewers so I only have a few points to raise.

72- This technique also relies on patients being able to take adequate photos of their throat and the view of their tonsils not being obscured by the tongue.

77- In the UK, the standard is to diagnose clinically rather than with a throat culture- this is rarely sent. Noted later in discussion 397 discusses circumstance with no access to lab diagnostics.

87- May not be necessary but could also consider the use of another commonly used tool - Feverpain

We thank the reviewers for their constructive feedback and suggestions in this iteration. In what follows, we respond to the last concerns raised and detail the changes that we made to improve our manuscript.

We modified the manuscript as follows:

- To enhance the motivation for supporting remote diagnosis of strep throat with AI, we incorporated additional factors that matter for developing (input quality validation) and measuring the success of such intervention (diagnostic accuracy and management of patients).
- To clarify what we achieved with our study and point toward future directions, in the introduction we now emphasize our goal for this study providing a stronger rationale for testing only correct recommendations, further pointing to opportunities for future investigations to remedy the limitations arising from this design choice.

In the remainder of the response letter, we use black bold font to state reviewers' comments. We use black font for our answers. We also attached text fragments of the changes made to the main manuscript in blue color.

Reviewer #2

I believe you have addressed the main points I raised. Of course the main limitations, which you seem to agree with, remain, but those are now acknowledged and the points of contention have been clarified.

A: We thank the reviewer for the overall positive assessment of our response. To further contend with the main limitation of our study, we further offer the following rationale and improvements to the manuscript. Because our study considered the evaluation of human behavior with a novel AI-based CDSS solution, our aim was to identify potential AI assistance elements that promote trust and understanding. In this context, we formulated an experimental design that allowed us to measure the effects of different explainable AI elements, even in the presence of some limitations. We stated this in the last paragraph in the introduction:

“In particular, our focus was in the influence of using alternate explainable AI elements in a CDSS on clinicians accuracy in predicting strep throat and which transparency techniques are associated with improved trust and understanding of the AI system.”

The HCI approach is acceptable, though I still believe that, for example, assuming infallible AI is a bit too much of a simplification. I am sure there will be scope for further work that will relax this and other assumptions.

A: We generally agree that assuming infallible AI, depending on the study's goals, can be too strong of a simplification, which we acknowledge as a limitation in the manuscript. Due to the primary goal of this study, however, we opted to not introduce this additional source of variability. This is because adding the accuracy of the model as an experimental factor creates new considerations that require careful study design. For instance, participants would need to

process considerably more samples to expose them to various model error modes, thus potentially resulting in participant fatigue or opt-outs. Further, exposing participants to AI errors at the beginning or end of the interaction can affect the trust they develop, depending on their expertise [1]. Our evidence suggests that even under a highly accurate AI, participants are not yet trusting recommendations that are beneficial to be followed in order to achieve a higher performance than they would when working with the Centor score. We agree that further studies should be conducted to evaluate how trust and adoption of AI recommendations is affected under different types of errors, i.e., false positives and false negatives.

We modified the manuscript as follows in the limitations paragraph:

“We recommend future works to explore how different types of AI errors and their presentation affect trust and adoption of advice in the presence of explanations that promote trust and understanding.”

[1] Nourani, M., King, J., & Ragan, E. (2020, October). The role of domain expertise in user trust and the impact of first impressions with intelligent systems. In *Proceedings of the AAAI Conference on Human Computation and Crowdsourcing* (Vol. 8, pp. 112-121).

Reviewer #3

In thinking about the practicality of this model in terms of diagnosis of strep A infections without the need for face to face consultation, I think we need to consider the practicalities of not only diagnosis but also risk management and identifying sick patients. I feel as a tool to fully manage a patient remotely in a safe manner, it would need to be considered how risk is managed i.e. how observations could be measured.

A: We agree that the implications of telemedicine go beyond diagnosis and should account for how to manage risk and patients who are actually sick and need medical attention, whether for strep or other conditions. We attempted to capture how risk is managed by including follow-up action recommendations questions in the survey (patients with higher risk of strep should take antibiotics or verifying the state of some patients who may be sick). We clarify the purpose of this measurement in the methods section:

“In addition to measuring diagnostic decisions, this additional question allowed us to assess how participants would manage sick patients who actually need antibiotic treatment and cases to be treated with symptomatic relief.”

72- This technique also relies on patients being able to take adequate photos of their throat and the view of their tonsils not being obscured by the tongue.

A: Indeed, for the algorithm to work patients must be able to acquire an adequate image of their throat. This is a technical challenge that can be overcome by careful smartphone application design that assists the patient or associate to acquire and upload high-quality throat images or videos. By providing feedback and guidance, on how to adjust the camera for example, the

patient or associates should be able to provide an adequate photo. An input quality assessment step can verify the adequacy of the image for the next diagnostic stage. These are algorithms we are working on, but felt would be outside the scope of this specific study. We have now listed this requirement in the introduction:

“Importantly, patients should provide adequate photos for both algorithmic and human processing, which can be achieved with additional guidance, feedback, and verification.”

77- In the UK, the standard is to diagnose clinically rather than with a throat culture- this is rarely sent. Noted later in discussion 397 discusses circumstance with no access to lab diagnostics.

A: We acknowledge the different conditions in which strep diagnosis can take place, including remote diagnosis where culture tests are simply not available, or protocols that do not involve culture tests. We have modified the introduction to capture this circumstance:

“Meanwhile, there are other protocols where it is common practice to diagnose strep throat clinically without sending throat cultures. Because an evaluation through telehealth services generally lacks point-of-care diagnostic tests, telehealth in its current form is a suboptimal option for patients with pharyngitis symptoms.”

“Due to the challenges associated with screening for strep throat in a remote setting without point-of-care tests and to support clinicians operating without medical tests, the aim of this study was to evaluate the use of an AI-based CDSS during telehealth services to identify strep throat from images of the throat.”

And the sentence mentioned in the discussion:

“In the context of remote strep throat screening with no access to laboratory tests or healthcare professionals operating without any form of medical tests”

87- May not be necessary but could also consider the use of another commonly used tool - Feverpain

A: We have listed Fever-PAIN as another clinical prediction rule when we refer to the Centor Score. Previous studies have compared both rules and there is evidence for similar performance of the rules, but some studies have identified lower sensitivity of Fever-PAIN scores [1]. We selected the Centor score due to its common use and potential familiarity of the study participants. Fever-PAIN was more recently developed and does not include the age factor, which matters in the carrier rates of the disease [2]. We now mention this in the introduction:

“the Modified Centor score includes age as a risk factor [16, 17], unlike other rules more recently proposed, such as Fever-PAIN [18].”

[1] Seeley, A., Fanshawe, T., Voysey, M., Hay, A., Moore, M., & Hayward, G. (2021). Diagnostic accuracy of Fever-PAIN and Centor criteria for bacterial throat infection in adults with sore throat: a secondary analysis of a randomised controlled trial. *BJGP open*, 5(6).

[2] Mitchell, M. S., Sorrentino, A., & Centor, R. M. (2011). Adolescent pharyngitis: a review of bacterial causes. *Clinical Pediatrics*, 50(12), 1091-1095.